# A BIOLOGICALLY-PLAUSIBLE ALTERNATIVE TO BACK-PROPAGATION USING PSEUDOINVERSE FEEDBACK CONNECTIONS

## ABSTRACT

Despite its successes in both practical machine learning and neural modeling, the backpropagation algorithm has long been considered biologically implausible (Crick, 1989). Previous solutions to this biological implausibility have proposed the existence of a separate, error feedback network, in which error at the final layer may be propagated backwards to earlier layers in a manner similar to backpropagation. However, biological evidence suggests that feedback connections in the cortex may function more similarly to an autoencoder, rather than being exclusively used as error feedback (Marino, 2020; Chen et al., 2024). Here, we attempt to unify these two paradigms by showing how autoencoder-like, inverse feedback connections may be used to minimize error throughout a feedforward neural network. Our proposed mechanism, Reciprocal Feedback, consists of two contributions: first we show how a modification of the Recirculation algorithm (Hinton & McClelland, 1988) is capable of learning the Moore-Penrose pseudoinverse of a pair of network weights. Then, we will show how, using a Newton-like method (Hildebrandt & Graves, 1927), locally-learned pseudoinverse feedback connections may be used to facilitate an alternative optimization method to traditional gradient descent - while alleviating the need to compute the weight transpose, or use direct feedback connections from the final layer. In the MNIST and CIFAR-10 classification tasks, our method obtains an asymptotic error similar to backpropagation, in fewer iterations than comparable biologically-plausible algorithms, such as Feedback Alignment (Lillicrap et al., 2014).

## 1 INTRODUCTION

The backpropagation algorithm is commonly regarded as the standard optimization algorithm for neural networks in machine learning (Rumelhart et al., 1986). Furthermore, artificial neural networks trained using the backpropagation algorithm have been shown to recapitulate features of biological networks, such as in the hippocampus (Chen et al., 2024) and visual cortex (Yamins et al., 2014). However, there are several aspects of the backpropagation algorithm which make it an unlikely candidate to be physically implemented in the brain (Crick, 1989) - one of these being the weight-transport problem (Grossberg, 1987). Our goal here is to explore network architectures and local learning rules that approximate the performance of backpropagation.

During the computation of the gradient in the backpropagation algorithm, the error at the final layer is passed "backwards" through the transpose of the each layer weight matrix, in order to compute the error with respect to each layer parameter. While computing the transpose of each weight matrix is fast and simple *in-silico*, there is no known biological mechanism by which error gradient signals may be precisely propagated backwards in biological neurons at a timescale relevant to learning.

Many algorithms have been proposed to circumvent the weight transport problem and approximate backpropagation in a biologically-plausible way. Several successful approaches have employed a separate error network of feedback connections to transmit final-layer error backwards, in a similar procedure to backpropagation (Lillicrap et al., 2014). Some of these feedback methods, including the weight-mirror (Akrout et al., 2019) and sign-symmetry (Xiao et al., 2018) method, have been shown to have the capacity to scale to large, complex datasets.

However, empirical evidence does not suggest that the physical bidirectional feedback connections which exist in the cortex are exclusively used to transmit error from the highest layer. Instead, it is likely that cortical feedback connections may be used to reconstruct activity patterns at lower layers using information from higher layers, and compute error signals locally (Mumford, 1992). Recent experimental work also supports the hypothesis that areas lower in the cortical processing hierarchy reconstruct activity patterns based on information from higher areas, particularly during memory tasks (Linde-Domingo et al. (2019), Favila et al. (2022), Naya et al. (2001)). This framework of predictive coding suggests that the cortical processing hierarchy may be more appropriately modeled by a series of stacked, autoencoder-like structures (Marino, 2020).

In this paper, we attempt to unify the experimentally-supported theory of the cortex as a series of autoencoders, with the error feedback models which have shown to be successful in emulating back-propagation and solving machine learning problems. *Specifically, we show how autoencoder-like, pseudoinverse feedback connections can be used to globally minimize error in a fully-connected, feedforward network.*

First, we will show how a simple modification of the Recirculation algorithm Hinton & McClelland (1988) is capable of training a pair of matrices to be the unique pseudoinverses of each other. Then, we will show that using only the pseudoinverse at each layer, it is possible to propagate the error backwards through the network, and adjust each layer weight in a direction that reduces the global error. For that derivation, we will apply the Newton-like method described by Hildebrandt & Graves (1927) and Ben-Israel (1966) as an alternative to gradient descent.

These two phases may be synchronized in a "wake-sleep" cycle (Figure 3). During the wake phase, predictions are generated from inputs, compared with targets, and forward weights are modified to minimize the error between them. During the "sleep" phase, random noise is generated at each layer, so that the forward and feedback matrices align to be the pseudoinverses of each other (Figure 2). Additionally, we also consider a parallel version of Reciprocal Feedback, in which feedback matrices are trained concurrently with forward matrices. We also show that our method outperforms the Feedback Alignment algorithm on the MNIST and CIFAR-10 image classification tasks.

## 2 RELATED WORK

### 2.1 RELATIONSHIP WITH TARGET-PROPAGATION ALGORITHMS

Using a series of stacked, locally-learned autoencoders to propagate error has previously been explored under the Target Propagation framework (Bengio, 2014). Under this framework, the forward pass of each layer, $f_l$ is defined by:

$$h_l = \sigma_i(W_l h_{l-1}) := f_i(h_{l-1})$$

The target at the final layer, $\hat{h}_L$, is then propagated backwards through layer-wise inverse functions, $g_l$, to provide a local target for each layer:

$$\hat{h}_l = \sigma_l(Q_l \hat{h}_{l+1}) := g_i(\hat{h}_{l+1})$$

These targets are used to update the forward weights, in a way that minimizes the local layer loss: $\mathcal{L}_l = ||h_l - \hat{h}_l||_2^2$.

Concurrently, feedback weights are trained as shallow autoencoders, with the loss function defined as:

$$\mathcal{L}_l = ||g_l(f_{l+1}(h_l)) - h_l||_2^2$$

In the case that the network is fully invertible, and each layer-wise inverse function is exact (i.e. $g_l(\hat{h}_{l+1}) = W_{l+1}^{-1}\sigma_{l+1}^{-1}(\hat{h}_{l+1})$), the target update is exactly that given by Gauss-Newton optimization, as proved in Meulemans et al. (2020).

More explicitly, since the exact inverse is factorizable, the optimal target direction (with respect to the final-layer error $e$, and learning rate $\lambda$) is approximately:

$$\Delta h_l \propto \hat{h}_l - h_l \approx \lambda \left( \prod_{k=l+1}^{L} J_k^{-1} \right) e$$

$$= \lambda \left( \prod_{k=L}^{l+1} J_k \right)^{-1} e$$

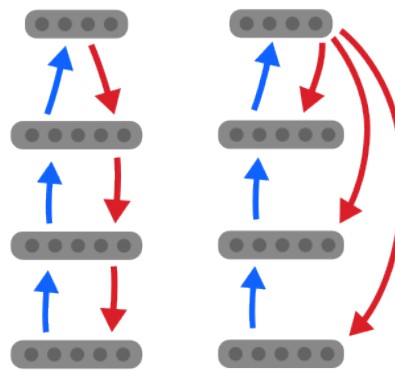

Figure 1: Comparison between the architecture proposed by Meulemans et al. (2020) (Right) and our reciprocal feedback architecture (Left).

However, as also recognized by Meulemans et al. (2020), in the case of non-invertible networks (in which feedback weights are interpreted to be the pseudoinverse instead) this does not necessarily hold, since the pseudoinverse is not generally factorizable in the same way as the exact inverse or transpose. In other words, $\left( \prod_{k=l}^{L} J_k^+ \right) \neq \left( \prod_{k=L}^{l} J_k \right)^+$. Since most feedforward neural network architectures in use are non-invertible, this is a significant limitation.

Meulemans et al. (2020) previously proposed adding additional feedback connections directly from the final output layer (Figure 1, left), which could be trained to directly learn the pseudoinverse of the Jacobian of the forward computation from layer $l$ to $L$. This was shown to be experimentally successful in similar classification tasks, but at the cost of sacrificing the modularity, locality, and biological realism underlying the original target propagation algorithm.

In this work, we will show that although the property of being the unique Moore-Penrose pseudoinverse is not preserved through matrix multiplication, the property of being a left reciprocal is (through preservation of pseudoinverse properties $\{1, 2, 3\}$).

Furthermore, drawing from the work of Ben-Israel on generalized inverses (Ben-Israel & Greville, 2003), we show that our recursively-defined, reciprocal feedback connections are sufficient to locally implement a Newton-like optimization method capable of minimizing the global error across the network. Overall, our work may serve as another, different theoretical framework by which Target Propagation algorithms may be understood.

## 2.2 Relationship with other weight-transport algorithms

Another biologically-plausible algorithm which uses random noise during a "sleep" period to learn appropriate feedback weights is the weight-mirror algorithm Akrout et al. (2019). In that case, the feedback weights are trained to approximate the transpose of the forward weights - as opposed to the pseudoinverse. This allows for a better direct approximation to backpropagation, but does not align with the autoencoder-like structure of feedback connections that would be expected.

## 3 Local learning of the pseudoinverse at each layer

We will first show that a linear modification of the Recirculation algorithm (Figure 2B and C) is capable of learning a pair of pseudoinverse matrices when its inputs are random, mean-zero noise. For generality, we will denote the forward (input-to-hidden) weight matrix as $V$, and backward (hidden-to-input) weight matrix as $U$. Furthermore, we will denote the random input layer vectors as $y$, and hidden layer vectors as $h$. We will also assume the random input vector $y$ has zero mean, uncorrelated elements and a variance of $\sigma^2$ for each element, thus $\mathbb{E}[yy^T] = \sigma^2 I$ (derivation in section A.3 of the Appendix).

To review, the Moore-Penrose pseudoinverse of $A$ is the unique matrix $A^+$ which satisfies the four Moore-Penrose conditions:

$$1. AA^+A = A \quad 2. A^+AA^+ = A^+ \quad 3. (AA^+)^T = AA^+ \quad 4. (A^+A)^T = A^+A$$

A theorem described in Ben-Israel & Cohen (1966) defines an iterative matrix computation for the pseudoinverse for an arbitrary matrix $\boldsymbol{A}$. Starting from a matrix $\boldsymbol{X}_0$ which satisfies $(\boldsymbol{X}_0\boldsymbol{A})^T = \boldsymbol{X}_0\boldsymbol{A}$, a sequence $\{\boldsymbol{X}_t\}$ is generated by:

$$\boldsymbol{X}_{t+1} - \boldsymbol{X}_t = \boldsymbol{X}_t - \boldsymbol{X}_t\boldsymbol{A}\boldsymbol{X}_t$$

where $\boldsymbol{X}_t$ converges to $\boldsymbol{A}^+$.

The pseudoinverse iteration above can be rewritten using the forward and feedback matrices $\boldsymbol{U}$ and $\boldsymbol{V}$, and the fact that $\mathbb{E}[\boldsymbol{y}\boldsymbol{y}^T] = \sigma^2\boldsymbol{I}$.

$$\Delta\boldsymbol{U} = \boldsymbol{U} - \boldsymbol{U}\boldsymbol{V}\boldsymbol{U} = \frac{1}{\sigma^2}(\boldsymbol{U} - \boldsymbol{U}\boldsymbol{V}\boldsymbol{U})\mathbb{E}[\boldsymbol{y}\boldsymbol{y}^T] = \frac{1}{\sigma^2}\mathbb{E}[(\boldsymbol{U}\boldsymbol{y} - \boldsymbol{U}\boldsymbol{V}\boldsymbol{U}\boldsymbol{y})\boldsymbol{y}^T]$$

Now, using similar dynamics to the Recirculation algorithm, we define $\boldsymbol{h}, \hat{\boldsymbol{y}}, \hat{\boldsymbol{h}}$ by:

$$\boldsymbol{h} = \boldsymbol{U}\boldsymbol{y}, \quad \hat{\boldsymbol{y}} = \boldsymbol{V}\boldsymbol{h}, \quad \hat{\boldsymbol{h}} = \boldsymbol{U}\hat{\boldsymbol{y}}$$

Altogether, the $\Delta\boldsymbol{U}$ above can be written in Recirculation dynamics as: $\Delta\boldsymbol{U} = \frac{1}{\sigma^2}\mathbb{E}[(\boldsymbol{h} - \hat{\boldsymbol{h}})\boldsymbol{y}^T]$.

So, if the input patterns $\boldsymbol{y}$ have a mean of $\boldsymbol{0}$, $\boldsymbol{U}$ will converge to the pseudoinverse of $\boldsymbol{V}$ when averaged over many inputs. Likewise, the same procedure on $\boldsymbol{V}$ will converge to the pseudoinverse of $\boldsymbol{U}$ (Figure 2A).

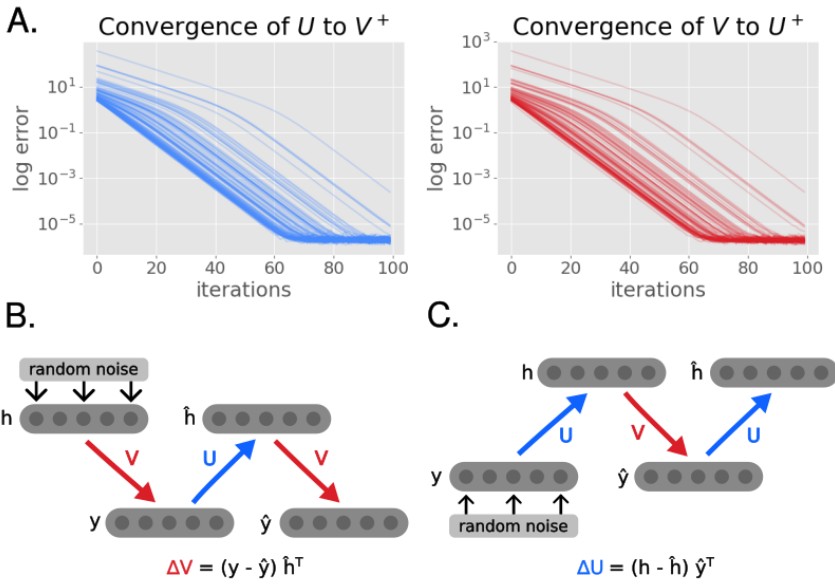

Figure 2: Using a local learning rule, the matrices $U$ and $V$ align to become the pseudoinverse of each other. A. The Frobenius norm of the difference between the pseudoinverse of $U$ (derived through its SVD) and $V$, and vice versa, over 100 random initializations. B. and C. are figures adapted from Hinton & McClelland (1988), showing the Recirculation learning procedure with random noise inputs.

Running this "sleep" phase algorithm on both $U$ and $V$ results in convergence of weights towards the pseudoinverse of each other, even with random initialization conditions. (implementation details in section A.3)

### 3.1 A PARALLEL MODIFICATION

To reduce the computational burden of re-calculating the exact pseudoinverse during the "sleep" phase, an alternative training scheme was used where the feedback weights were trained concurrently with the forward weights.

These are done using a simplified linear autoencoder learning rule, similar to feedback learning rule in our modified recirculation learning procedure:

$$\delta_{\boldsymbol{W}^+} = (\boldsymbol{W}^+\boldsymbol{W}\boldsymbol{y} - \boldsymbol{y})(\boldsymbol{W}\boldsymbol{y})^T, \quad \Delta\boldsymbol{W} = -\lambda\delta_{\boldsymbol{W}^+}$$

However, since the inputs, $y$, are not guaranteed to be mean-zero white noise, the learned feedback weights may not be the exact pseudoinverse.

## 4 TRAINING THE WHOLE MULTI-LAYER NETWORK

The network architecture we will consider in this paper is a fully-connected, feedforward network consisting of a series of hidden layers, connected by forward weight matrices ($\boldsymbol{W}_l$), and feedback weight matrices ($\boldsymbol{W}_l^+$).

As in a standard feedforward network, each layer consists of an application of matrix-multiplication to the previous hidden layer activation vector, followed by the application of a bijective, nonlinear activation function ($\sigma(\cdot)$) on each element. Mathematically, for a network consisting of $L$ layers, the pre-activation vector ($\boldsymbol{a}_l$) and activation vector ($\boldsymbol{h}_l$) are defined by:

$$\boldsymbol{h}_0 = \boldsymbol{x}, \quad \boldsymbol{a}_l = \boldsymbol{W}_l\boldsymbol{h}_{l-1}, \quad \boldsymbol{h}_l = \sigma(\boldsymbol{a}_l)$$

for each layer $l \leq L$.

We will define the *scalar* error function $\mathcal{E}$ as the mean squared value of the difference between the final-layer output ($\boldsymbol{h}_L$), and target output ($\boldsymbol{h}_L^*$). We will also define the error *vector* $\boldsymbol{e}$ as the difference between the final layer activation ($\boldsymbol{h}_L$) and the final layer target ($\boldsymbol{h}_L^*$):

$$\mathcal{E} = \frac{1}{2}||\boldsymbol{h}_L - \boldsymbol{h}_L^*||_2^2, \quad \boldsymbol{e} = \boldsymbol{h}_L - \boldsymbol{h}_L^*$$

During gradient descent, the gradient of $\mathcal{E}$ with respect to each parameter $\boldsymbol{W}_l$ approaches 0: $(\boldsymbol{J}_{W_l}^{\mathcal{E}})^T\boldsymbol{e}(\boldsymbol{x}, \boldsymbol{W}_l) \to 0$. By optimizing $\boldsymbol{e}$ directly, such that $\boldsymbol{e}(x, W_l) \to 0$, the gradient with respect to $\mathcal{E}$ will also approach 0. In essence, minimizing the error vector $\boldsymbol{e}$ minimizes the scalar error $\mathcal{E}$.

### NOTATION AND PRELIMINARIES

Following the notation of Magnus & Neudecker (2019), the Jacobian of a vector $\boldsymbol{y}$ with respect to a matrix $\boldsymbol{X}$ is:

$$\boldsymbol{J}_X^y = \frac{\partial\boldsymbol{y}}{\partial(\text{vec }\boldsymbol{X})^T}$$

It is also relevant to note that the chain rule for Jacobians is applied in reverse order compared to the chain rule for gradients. For functions $Y(X)$ and $X(Z)$, the chain rule for Jacobians is $\boldsymbol{J}_Z^Y = \boldsymbol{J}_X^Y\boldsymbol{J}_Z^X$.

We will also assume that the vector function $\boldsymbol{e}$ is in the class $C'(X_0)$, meaning that has a continuous differential at every point $x \in X_0$. So, for each parameter $\boldsymbol{W}_l$, there exists a differential with Jacobian matrix $\boldsymbol{J}_{W_l}^E$ such that for two nearby $\boldsymbol{W}_l^0$ and $\boldsymbol{W}_l^1$,

$$E(\boldsymbol{x}, \boldsymbol{W}_l^1) - E(\boldsymbol{x}, \boldsymbol{W}_l^0) - \boldsymbol{J}_{W_l}^E(\text{vec }\boldsymbol{W}_l^1 - \text{vec }\boldsymbol{W}_l^0) = r(\text{vec }\boldsymbol{W}_l^0)||\text{vec }\boldsymbol{W}_l^1 - \text{vec }\boldsymbol{W}_l^0||$$

where $r$ is a function in which

$$\lim_{\text{vec }\boldsymbol{W}_l^1 \to \text{vec }\boldsymbol{W}_l^0}||r(\text{vec }\boldsymbol{W}_0)|| = 0$$

A Newton-like method described in Hildebrandt & Graves (1927) and Ben-Israel (1966) uses the left multiplicative reciprocal of the Jacobian of a function to minimize it. Unlike gradient descent, it does not require computing the transpose of the Jacobian - making it potentially a more biologically-plausible method for global error minimization throughout the network. Briefly, it can be stated as follows:

Figure 3: Illustration of the three phases of network operation in a simple 4-layer network. A. Forward propagation of input $x$, represented by green arrows. B. Backwards propagation of the error vector (represented by the orange arrows) through the feedback matrices. C. "Sleep" phase, where forward and feedback matrices are trained to be the pseudoinverse of each other. Part C. of Figure 1 illustrates this section in greater detail.

**Theorem 1** *If $X_0$ is a subset of $\mathbb{R}^n$, $\boldsymbol{y}_0 \in \mathbb{R}^k$ is a vector, $B_r(\boldsymbol{y}_0)$ is a ball of radius $r$ centered at $\boldsymbol{y}_0$, and $F : (X_0, B_r(\boldsymbol{y}_0)) \to \mathbb{R}^m$ is a vector function such that:*

1. *There exists a matrix $\boldsymbol{A} : B_r(y_0) \to \mathbb{R}^k$, with left reciprocal $\boldsymbol{T}$ and positive number $M$ such that:*

$$\boldsymbol{T}\boldsymbol{A}\boldsymbol{y} = \boldsymbol{y}$$
$$M||\boldsymbol{T}|| < 1$$
$$||\boldsymbol{A}(\boldsymbol{y}_1 - \boldsymbol{y}_2) - F(\boldsymbol{x}, \boldsymbol{y}_1) + F(\boldsymbol{x}, \boldsymbol{y}_2)|| \leq M||\boldsymbol{y}_1 - \boldsymbol{y}_2||$$

*for all $\boldsymbol{x} \in X_0$ and every $\boldsymbol{y} \in B_r(y_0)$ where $\boldsymbol{y} \in R(\boldsymbol{T})$.*

2. *the radius $r$ satisfies:*

$$||\boldsymbol{T}||||F(\boldsymbol{x}, \boldsymbol{y}_0)|| < r(1 - M||\boldsymbol{T}||)$$

*for all $\boldsymbol{x} \in X_0$*

*Then there exists a solution $\boldsymbol{y}^*$ which for every $\boldsymbol{x} \in X_0$ satisfies*

$$\boldsymbol{T}F(\boldsymbol{x}, \boldsymbol{y}^*) = 0$$

*which can be obtained using the iteration:*

$$\boldsymbol{y}_{t+1} = \boldsymbol{y}_t - \boldsymbol{T}F(\boldsymbol{x}, \boldsymbol{y}_t)$$

A proof of this theorem, based on that of Ben-Israel (1966), can be found in the Appendix section A.2.

In the context of an artificial neural network, we can consider $X_0$ to be the set of n-dimensional input vectors, $\boldsymbol{y}$ to be the parameters of the network (in vector form), and $F$ to be the function mapping each input and parameter pair to a vector (in our case, the error vector resulting from each input-target pair). $\boldsymbol{y}_0$ would be initial value of the parameter, and the solution $\boldsymbol{y}^*$ is the final value of the parameter at the end of that iteration. Since $\boldsymbol{y}$ is a vector, and $\boldsymbol{W}_l$ are each matrices, we will use a vectorized version of each weight matrix for our derivation: vec $\boldsymbol{W}_l$.

## 4.1 DERIVING THE JACOBIAN AT EACH LAYER

First, we will derive the Jacobian matrix, $\boldsymbol{J}_{W_l}^E$, of the error vector with respect to each parameter $\boldsymbol{W}_l$ in terms of the forward weight matrices and activation vectors.

Using the chain rule described above, the Jacobian matrix of each layer parameter is given by:

$$[J_{W_l}^E]_{ij} = \frac{\partial E}{\partial W_{ij}^l} = \sum_k \frac{\partial h_k^l}{\partial W_{ij}} \frac{\partial E}{\partial h_k^l} = a_j^{l-1} \frac{\partial E}{\partial h_i^l}$$

More compactly, in matrix notation:

$$\boldsymbol{J}_{W_l}^E = \boldsymbol{J}_{W_l}^{h_l} \boldsymbol{J}_{h_l}^E = (\boldsymbol{a}_{l-1} \otimes \boldsymbol{J}_{h_l}^E)$$

Next, we need to find a recursive expression to compute $\boldsymbol{J}_{h_l}^E$ at each layer. (similarly to how the backpropagation algorithm uses a recursive expression to compute the gradient $\nabla_{W_l} e$). Using the chain rule again:

$$\boldsymbol{J}_{h_l}^E = \boldsymbol{J}_{h_{l+1}}^E \boldsymbol{J}_{a_{l+1}}^{h_{l+1}} J_{h_l}^{a_{l+1}} = \boldsymbol{J}_{h_{l+1}}^E \mathcal{D}_\sigma \boldsymbol{W}_{l+1}$$

where $\mathcal{D}_\sigma$ is a diagonal matrix representing the derivative of the elementwise, nonlinear operator $\sigma(\cdot)$.

All of the derivations so far are the same as those used in backpropagation, but transposed (since we are computing the Jacobian rather than the gradient matrix). Now, we have a recursive expression for the Jacobian of each forward matrix parameter, $\boldsymbol{W}_l$.

## 4.2 FINDING RECIPROCALS OF THE JACOBIAN

In order to apply the Hildebrandt-Graves Theorem to optimize the parameters $\boldsymbol{W}_l$, we need to find left reciprocal matrices $\boldsymbol{\mathcal{B}}_l$, such that for each parameter, $\boldsymbol{\mathcal{B}}_l \boldsymbol{J}_{W_l}^E \boldsymbol{y} = \boldsymbol{y}$.

To aid in this derivation, we will also define the left reciprocal of the activation Jacobian ($\boldsymbol{J}_{h_l}^E$) as $\boldsymbol{B}_l$. Note the difference between $\boldsymbol{B}_l$ and $\boldsymbol{\mathcal{B}}_l$.

To begin finding a suitable reciprocal, we will assume that we have access to the Moore-Penrose pseudoinverse of each forward weight (as we've shown is obtainable with a modification of the Recirculation algorithm). Following standard notation, we will denote the unique Moore-Penrose pseudoinverse of each layer weight matrix as $\boldsymbol{W}_L^+$. Physically, would be as a feedback connection between adjacent layers.

First, we will find a left reciprocal of the activation Jacobian, $\boldsymbol{J}_{h_l}^E$. Like the Jacobian itself, the reciprocal $\boldsymbol{B}_l$ can be computed recursively, using the pseudoinverses of each layer weight matrix:

$$\boldsymbol{B}_l = (\mathcal{D}_\sigma \boldsymbol{W}_{l+1})^+ \boldsymbol{B}_{l+1} = \boldsymbol{W}_{l+1}^+ \mathcal{D}_\sigma^+ \boldsymbol{B}_{l+1}$$

By checking each of the four Moore-Penrose conditions (Penrose, 1955), it can be shown that this recursion on $\boldsymbol{B}_l$ preserves the pseudoinverse properties $\{1, 2, 3\}$. Furthermore, when every $\boldsymbol{W}_l$ is full row-rank, $\boldsymbol{B}_l$ acts as a multiplicative left reciprocal (Tian, 2020). And, if we initialize each $\boldsymbol{W}_l$ with random weights, it is almost certainly going to be a full-rank matrix.

Using the activation reciprocal, $\boldsymbol{B}_l$, the reciprocal of the layer parameter, $\boldsymbol{\mathcal{B}}_l$, can be defined with:

$$\boldsymbol{\mathcal{B}}_l = \left( \frac{1}{||\boldsymbol{a}_{l-1}||_2^2} \boldsymbol{a}_{l-1}^T \otimes \boldsymbol{B}_l \right)$$

So, $\mathcal{B}_l$ has the property that:

$$\mathcal{B}_l J_{W_l}^E = \left( \frac{1}{||\boldsymbol{a}||_2^2} \boldsymbol{a}_{l-1}^T \otimes \boldsymbol{B}_l \right) \left( \boldsymbol{a}_{l-1} \otimes J_{h_l}^E \right)$$

$$= \boldsymbol{B}_l J_{h_l}^E$$

This would imply that for all $\boldsymbol{y}$ such that $\boldsymbol{B}_l J_{h_l}^E \boldsymbol{y} = \boldsymbol{y}$, then $\mathcal{B}_l J_{W_l}^E \boldsymbol{y} = \boldsymbol{y}$.

Therefore, using the Jacobian $J_{W_l}^E$ as the matrix $\boldsymbol{A}$, and the reciprocal $\mathcal{B}_l$ as its reciprocal $\boldsymbol{T}$, Theorem 1 allows an update direction to be computed for $\boldsymbol{W}_l$ at each layer which minimizes the error vector.

Specifically, the weight update direction $\delta_{W_l}^t$ is given by:

$$\delta_{W_l} = \mathcal{B}_l \boldsymbol{e}(x, W_l)$$

$$= \left( \frac{1}{||\boldsymbol{a}||_2^2} \boldsymbol{a}_{l-1}^T \otimes \boldsymbol{B}_l \right) \boldsymbol{e}(x, W_l)$$

$$= \frac{1}{||\boldsymbol{a}||_2^2} \boldsymbol{B}_l \boldsymbol{e}(x, W_l) \boldsymbol{a}_{l-1}^T$$

And during each step of learning,

$$\Delta \boldsymbol{W}_l = -\lambda \delta_{W_l}$$

Where $\lambda$ is the manually determined learning rate.

In conclusion, Theorem 1 allows us to minimize the error vector $\boldsymbol{e}$ with respect to each parameter $\boldsymbol{W}_l$. Unlike backpropagation, we only require the pseudoinverse of each forward weight matrix, rather than its exact transpose, making it potentially more biologically-plausible by circumventing the weight transpose problem.

However, to restate some important assumptions we have made, in order for Theorem 1 to minimize the error of the network:

1. The condition number of the Jacobian and reciprocal must be kept low, so that condition 1 can be satisfied.

2. Each weight matrix $W_l$ must be full-row rank for the matrix $\mathcal{B}_l$, as defined above, to be its left reciprocal. However, if each weight matrix is initialized from a random distribution, it is almost certainly going to be full-rank.

3. The nonlinear operator $\sigma$ must be bijective and continuous. For instance, this could be a leaky softplus activation.

## 5 COMPUTATIONAL EXPERIMENTS

This algorithm was tested on two classification problems - MNIST and CIFAR-10. For both datasets, we used the cross-entropy loss functionand SGD with a fixed batch size. Hyperparameter details are given in the Appendix. However, we used a noncontinuous, leaky ReLU nonlinearity in order to reduce computational costs. Implementation and training was done using the PyTorch library (Paszke et al., 2019).

We benchmarked our algorithm against the biologically-plausible Feedback Alignment (FA) algorithm, as described in Lillicrap et al. (2014). Feedback Alignment uses fixed, random feedback connections to propagate error backwards throughout the network, in place of the weight transpose (as in backpropagation). Code is available in the supplementary material.

### 5.1 ARCHITECTURE AND IMPLEMENTAION DETAILS

The MNIST classifier had 5 hidden layers, and an architecture of $(28 \times 28) - 400 - 200 - 100 - 50 - 10$, and the CIFAR-10 classifier has 4 hidden layers: $(32 \times 32 \times 3) - 1000 - 500 - 100 - 10$.

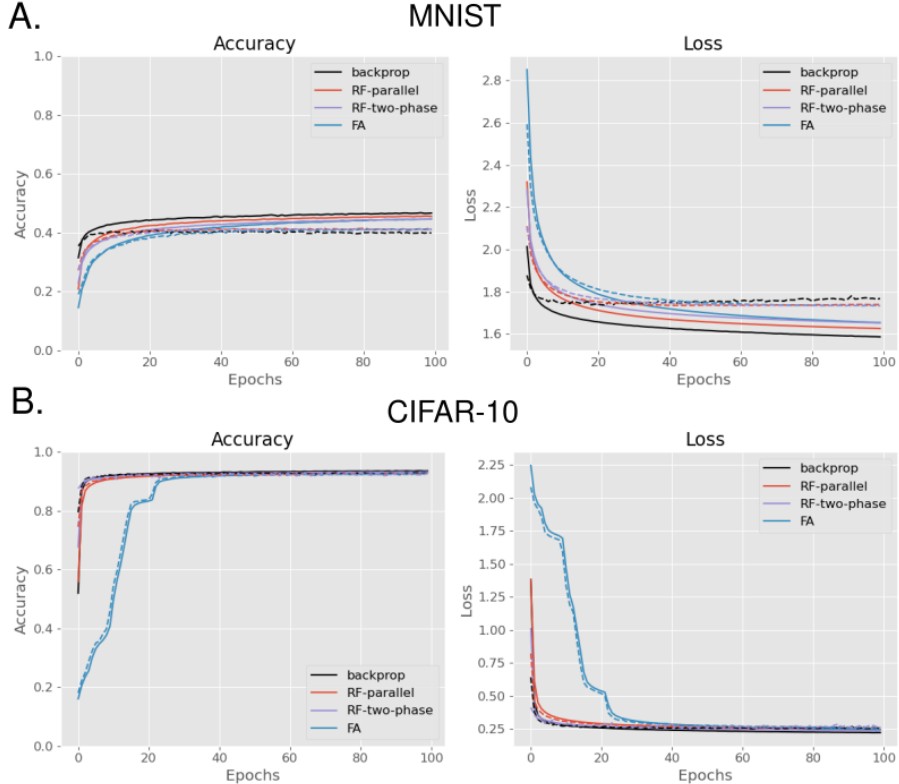

Figure 4: Backpropagation (backprop), Two-phase Reciprocal Feedback (RF-two-phase), Parallel Reciprocal Feedback (RF-parallel) and Feedback Alignment (FA), on A. MNIST digit classification and B. CIFAR-10 natural image classification datasets. Solid lines indicate training loss and accuracy, while dashed lines indicate test loss and accuracy.

It should be noted that CIFAR-10 is generally not solved well with a fully-connected, feedforward network, and requires a locally-connected architecture to achieve high accuracy (Lee et al., 2015).

Each algorithm was initialized with 20 random seed and hyperparameter configurations. Those with the highest test accuracy are reported. The optimal hyperparameters are listed in Section A.1.

Pseudocode implementations of two-phase and parallel Reciprocal Feedback are included in Section A.6.

| Algorithm | MNIST | | CIFAR-10 | |
|---|---|---|---|---|
| | **Accuracy (%)** | **Epochs** | **Accuracy (%)** | **Epochs** |
| Backpropagation | 92.99 | 42 | 40.76 | 16 |
| Reciprocal Feedback (Parallel) | 92.65 | 79 | 41.46 | 44 |
| Reciprocal Feedback (Two-phase) | 92.81 | 64 | 41.34 | 42 |
| Feedback Alignment | 92.67 | 71 | 40.29 | 80 |

Table 1: Final classification accuracy, and number of epochs required to reach the 90% percentile of asymptotic error for each algorithm

## 6 DISCUSSION

We have shown how locally-learned pseudoinverse feedback connections can be used to train a feedforward, fully-connected neural network, using an alternative optimization method to gradient descent. With our method, we alleviate the need to compute the weight transpose at each layer - suggesting a possible solution to the weight transport problem of the backpropagation algorithm. Furthermore, the use of pseudoinverse feedback connections may better align with the evidence-based, neuroscientific model of the cortex as a series of autoencoder-like structures.

To apply Theorem 1 to a neural network, any suitable left multiplicative reciprocal of the Jacobian may be used, not necessarily the $\mathcal{B}$ recursion based on the Moore-Penrose pseudoinverse. Single-layer, autoencoder-like algorithms other than Recirculation may also be able to learn a suitable generalized inverse at each layer. For instance, Tapson & van Schaik (2013) describes an online, biologically-plausible algorithm for computing the pseudoinverse of a network's weights. Furthermore, given the rank-one, outer-product structure of forward matrix updates, it may be possible to make local, online updates to the exact pseudoinverse using the Sherman-Morrison formula.

This method may also be related to Gauss-Newton optimization for neural networks, which uses the direct pseudoinverse of the whole network Jacobian at each layer (Botev et al., 2017). The main issue with implementing this using only layer-wise pseudoinverses is that unique, direct pseudoinverse of the whole network cannot be composed sequentially using each layer's pseudoinverse. In other words, since it is an approximation of the second-derivative, the chain rule for Hessians must be used instead, followed by taking the pseudoinverse of the result separately (see section A.4). The optimization method in Theorem 1 is understudied in the context of machine learning, and its convergence properties compared to gradient descent are still unknown.

Due to the cost of computing and updating the pseudoinverse in a digital computer, this method is more computationally expensive than backpropagation. However, when implemented in a physical, analog system, the relative costs of computing the inverse and transpose may be different. In the context of neural modeling, the weights of artificial neural networks are often considered analogous to resistance parameters of physical neuron models (Abbott & Dayan, 2005). Because of the applicability of Ohm's and Kirchoff's laws, analogously representing quantities as resistances have allowed certain matrix computations (such as matrix inversion) to be performed with greater efficiency than in digital computers (Sun et al., 2019).

The learning algorithms analyzed here in a feedforward model can potentially be generalized to stacked recurrent neural networks, such as a predictive autoencoder with a recurrent hidden layer trained to learn sequences of inputs (Chen et al., 2024).

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

# A  APPENDIX

## A.1  HYPERPARAMETER DETAILS

Hyperparameters were chosen using the HyperOpt search algorithm, with the search domains listed below. ASHA scheduler was used to optimize hyperparameter searches, with a maximum number of epochs of 100, such that poorly performing hyperparameter initializations were stopped early. The best result out of 20 random configurations were used for each algorithm.

For the two-phase Reciprocal Feedback implementation, pseudoinverses were recalculated once every 20 update steps.

| Hyperparameter | Search Domain |
|---|---|
| Forward learning rate | $[10^{-4}, 10^{-2}]$ |
| Feedback learning rate | $[10^{-6}, 10^{-3}]$ |
| Batch Size | 32 |
| Epochs | $[1, 100]$ |

Table 2: Hyperparameter search spaces for all algorithms.

| | Backprop | FA | RF (parallel) | RF (two-phase) |
|---|---|---|---|---|
| Forward learning rate | 0.0020118 | 0.00084333 | 0.0016435 | 0.004915 |
| Feedback learning rate | - | - | 0.000075683 | - |

Table 3: Optimal hyperparameters for MNIST

| | Backprop | FA | RF (parallel) | RF (two-phase) |
|---|---|---|---|---|
| Forward learning rate | 0.000666 | 0.00030816 | 0.00056066 | 0.000447 |
| Feedback learning rate | - | - | 0.00048527 | - |

Table 4: Optimal hyperparameters for CIFAR-10.

## A.2 PROOF OF THEOREM 1

This proof of Theorem 1 is based on that in the original paper Hildebrandt & Graves (1927), and its extension to pseudoinverses in Ben-Israel (1966).

For this section, we will consider the vector space $E^n$, equipped with the vector 2-norm, and the matrix space $E^{n \times m}$ with the spectral norm. Together, these have the property that for any $x \in E^n$ and $A \in E^{n \times m}$, $||Ax|| \leq ||A|| ||x||$.

Furthermore, let the function $F$ be in the class $C'(Y)$. (i.e. both the mapping $F$ and its Jacobian are continuous in the open set $Y \in E^n$). Also, we will notate the open ball centered at $x_0$ as $B_r(x_0) = \{x \in E^n : ||x - x_0|| < r\}$.

Theorem 1 states the following: let $X_0$ be a subset of $E^n$, $y_0 \in E^n$ be a point, and $F : X_0 \times B_r(y_0) \to E^m$ be a function such that:

1. there exists a linear function $A : B_r(y_0) \to E^m$, with reciprocal $T$ and positive number $M$ such that:

$$TAy = y$$
$$M||T|| < 1$$
$$||A(y_1 - y_2) - F(x, y_1) + F(x, y_2)|| \leq M||y_1 - y_2||$$

   for all $x \in X_0$ and every $y \in B_r(y_0)$ where $y \in R(T)$.

2. the radius $r$ satisfies:

$$||T|| ||F(x, y_0)|| < r(1 - M||T||)$$

   for all $x \in X_0$

Then there exists a solution $y(x)$ which for every $x \in X_0$ satisfies

$$TF(x, y(x)) = 0$$

Consider the function:

$$G(x, y) = y - TF(x, y)$$

where $y \in B_r(y_0) \bigcap R(T)$. Then, using the property of the reciprocal,

$$
\begin{aligned}
G(x, y_1) - G(x, y_2) &= y_1 - y_2 - TF(x, y_1) + TF(x, y_2) \\
&= TA(y_1 - y_2) - TF(x, y_1) + TF(x, y_2) \\
&= T(A(y_1 - y_2) - F(x, y_1) + F(x, y_2))
\end{aligned}
$$

So, by assumption 1:

$$||G(x, y_1) - G(x, y_2)|| \leq ||T|| M ||y_1 - y_2|| \qquad < ||y_1 - y_2||$$

So, for any $y \in B_r(y_0)$, $G(x, y) \in B_r(G(x, y_0))$.

Next, consider the sequence $\{y_k\}$ defined by:

$$y_1 = G(x, y_0)$$
$$y_{k+1} = G(x, y_k)$$

Using assumption 2:

$$||y_1 - y_0|| = ||y_0 - TF(x, y_0) - y_0||$$
$$\leq ||T||||F(x, y_0)||$$
$$< r(1 - M||T||)$$

Define $p = M||T|| < 1$. Then, by induction, we can show that $\{y_k\}$ is Cauchy:

$$||y_{k+1} - y_k|| < p^k r(1 - p)$$

Furthermore, it can also be shown by induction that $y_{k+1} - y_k \in R(T)$.

So, $\{y_k\}$ is convergent in $B_r(y_0) \bigcap R(T)$, and converges towards a vector $y^*$, which satisfies

$$TF(x, y^*) = 0$$

And, if $N(T) \subset N(A^T)$, it also satisfies:

$$A^T F(x, y^*) = 0$$

Meaning that the minima that Theorem 1 converges to would be the same as that as gradient descent.

### A.3 THE EXPECTED VALUE OF THE OUTER PRODUCT OF RANDOM, MEAN-ZERO VECTORS

In this section, we will show that $\boldsymbol{E}[\boldsymbol{y}\boldsymbol{y}^T] = \sigma^2 \boldsymbol{I}$. Consider a random vector $\boldsymbol{y}$, whose elements are independent with a mean of 0 and variance of $\sigma^2$. Each of these vectors can be decomposed into the form:

$$\boldsymbol{y} = \sum_{i=1}^{N} y_i \boldsymbol{e}^{(i)}$$

where $\boldsymbol{e}^{(i)}$ is a one-hot vector at index $i$, and $y_i$ is a random scalar.

So, the outer product of two of these random vectors can be rewritten as:

$$\boldsymbol{y}\boldsymbol{y}^T = (\sum_{i=1}^{N} y_i \boldsymbol{e}^{(i)})(\sum_{j=1}^{N} y_j \boldsymbol{e}^{(j)T})$$
$$= \sum_{i=1}^{N} \sum_{j=1}^{N} y_i \boldsymbol{e}^{(i)} \boldsymbol{e}^{(j)T} y_j$$

Since $y$ is random with mean 0, when $i \neq j$ then $\mathbb{E}[y_i y_j] = \mathbb{E}[y_i]\mathbb{E}[y_j] = 0$. Otherwise, $\mathbb{E}[y_i^2] = \sigma^2$.

$$\mathbb{E}[\boldsymbol{y}\boldsymbol{y}^T] = \mathbb{E}[\sum_{i=1}^{N} \sum_{j=1}^{N} y_i \boldsymbol{e}^{(i)} \boldsymbol{e}^{(j)T} y_j]$$
$$= \sum_{i=1}^{N} \sum_{j=1}^{N} \mathbb{E}[y_i y_j] \boldsymbol{e}^{(i)} \boldsymbol{e}^{(j)T}$$
$$= \sum_{i=1}^{N} \mathbb{E}[y_i^2] \boldsymbol{e}^{(i)} \boldsymbol{e}^{(i)T}$$
$$= \sigma^2 \sum_{i=1}^{N} \boldsymbol{e}^{(i)} \boldsymbol{e}^{(i)T}$$
$$= \sigma^2 \boldsymbol{I}$$

### A.4 DETAILS OF PSEUDOINVERSE ITERATION

For the simulations in Figure 3, $U$ and $V$ were initialized as $(10 \times 10)$ square matrices. Their values were randomly initialized, with $\boldsymbol{E}_U$ and $\boldsymbol{E}_V$ randomly sampled from a uniform distribution with range $[-1, 1]$. This was done in order to keep the condition number low. Furthermore, initializations with a condition number over 20 were removed.

$$\boldsymbol{U}_0 := \boldsymbol{I} - \boldsymbol{E}_U$$
$$\boldsymbol{V}_0 := \boldsymbol{I} - \boldsymbol{E}_V$$

Each matrix was updated with the local learning rule:

$$\boldsymbol{U}_{t+1} := \lambda \boldsymbol{U}_t + (1 - \lambda)(\boldsymbol{U}_t - \boldsymbol{U}_t \boldsymbol{V}_t \boldsymbol{U}_t)$$
$$\boldsymbol{V}_{t+1} := \lambda \boldsymbol{V}_t + (1 - \lambda)(\boldsymbol{V}_t - \boldsymbol{V}_t \boldsymbol{U}_t \boldsymbol{V}_t)$$

Where $\lambda$ is a decay constant used to increase stability. During simulations this was set to $\lambda = 0.9$.

### A.5 COMPARISON WITH GAUSS-NEWTON OPTIMIZATION

Newton's optimization method involves preconditioning the gradient with the inverse of the Hessian, in order to account for the curvature of the loss landscape. Geometrically, this can be thought of as taking larger "steps" when the landscape is flatter, and smaller "steps" when it is sharper - generally resulting in fewer steps needed than regular gradient descent.

In the context of optimizing the weights of a neural network, with the Hessian $H_{W_l}$, the block-diagonal approximation of the optimal Newton direction is given by:

$$\delta_{W_l} = (H_{W_l})^{-1} \nabla_{W_l} e \tag{1}$$

The Gauss-Newton method approximates the Hessian with $H_{W_l} \approx (J_{W_l}^e)^T J_{W_l}^e$. Using the Gauss-Newton approximation, and expanding the gradient term in (1) results in:

$$\delta_{W_l} = ((J_{W_l}^e)^T J_{W_l}^e)^{-1} (J_{W_l}^e)^T e \tag{2}$$

Furthermore, if we assume $J_{W_l}^e$ has full row rank, (2) is equivalent to the pseudoinverse expression:

$$\delta_{W_l} = (J_{W_l}^e)^+ e \tag{3}$$

If it was the case that, like the transpose, $(J_{W_{l+1}}^e J_{W_l}^e \dots)^+ = \dots (J_{W_l}^e)^+ (J_{W_{l+1}}^e)^+$, then our method would be equivalent to block-diagonal Gauss-Newton optimization (Botev et al., 2017). Unfortunately, that does not generally apply to the pseudoinverse. Instead, the expected recursion with respect to the whole network would be

$$(J_{W_l}^e)^+ = (W_l^T \mathcal{D}_\sigma^T (J_{W_{l+1}}^e)^T J_{W_{l+1}}^e \mathcal{D}_\sigma W_l)^{-1} W_l^T \mathcal{D}_\sigma^T (J_{W_{l+1}}^e)^T \tag{4}$$

Which is significantly more difficult to implement using local connections.

## A.6 PSEUDOCODE FOR RF ALGORITHMS

---

**Algorithm 1** Two-phase Reciprocal Feedback

---

**for** $((\boldsymbol{x}, \boldsymbol{h}_L^*) \in$ batch **do**
  $\boldsymbol{W}_1^+, \ldots, \boldsymbol{W}_L^+ \leftarrow$ Update Pseudoinverse          ▷ (sleep)
  $\boldsymbol{h}_0 \leftarrow \boldsymbol{x}$
  **for** $l \in L$ **do**          ▷ Forward pass (wake)
    $\boldsymbol{a}_l \leftarrow \boldsymbol{W}_l \boldsymbol{h}_{l-1}$
    $\boldsymbol{h}_l \leftarrow \sigma(\boldsymbol{a}_l)$
  **end for**
  $\boldsymbol{e} \leftarrow \boldsymbol{h}_L - \boldsymbol{h}_L^*$
  **for** $l \in L$ **do**          ▷ Backward pass (wake)
    $\boldsymbol{B}_l \leftarrow \boldsymbol{W}_l^+ \boldsymbol{\mathcal{D}}_{\boldsymbol{\sigma}}^+ \boldsymbol{B}_{l+1}$
    $\delta_{W_l} \leftarrow (\boldsymbol{B}_l \boldsymbol{e}) \boldsymbol{a}_{l-1}^T$
  **end for**
  $\boldsymbol{W}_1, \ldots, \boldsymbol{W}_L \leftarrow \boldsymbol{W}_1 - \lambda \delta_{W_1}, \ldots, \boldsymbol{W}_L - \lambda \delta_{W_L}$
**end for**

---

---

**Algorithm 2** Parallel Reciprocal Feedback

---

**for** $((\boldsymbol{x}, \boldsymbol{h}_L^*) \in$ batch **do**
  $\boldsymbol{h}_0 \leftarrow \boldsymbol{x}$
  **for** $l \in L$ **do**          ▷ Forward pass
    $\boldsymbol{a}_l \leftarrow \boldsymbol{W}_l \boldsymbol{h}_{l-1}$
    $\boldsymbol{h}_l \leftarrow \sigma(\boldsymbol{a}_l)$
  **end for**
  $\boldsymbol{e} \leftarrow \boldsymbol{h}_L - \boldsymbol{h}_L^*$
  **for** $l \in L$ **do**          ▷ Backward pass
    $\boldsymbol{B}_l \leftarrow \boldsymbol{W}_l^+ \boldsymbol{\mathcal{D}}_{\boldsymbol{\sigma}}^+ \boldsymbol{B}_{l+1}$
    $\delta_{\boldsymbol{W}_l} \leftarrow (\boldsymbol{B}_l \boldsymbol{e}) \boldsymbol{a}_{l-1}^T$
    $\delta_{\boldsymbol{W}_l^+} \leftarrow (\boldsymbol{W}_l^+ \boldsymbol{W}_l \boldsymbol{a}_l - \boldsymbol{a}_l) \boldsymbol{a}_{l-1}^T$
  **end for**
  $\boldsymbol{W}_1, \ldots, \boldsymbol{W}_L \leftarrow \boldsymbol{W}_1 - \lambda \delta_{\boldsymbol{W}_1}, \ldots, \boldsymbol{W}_L - \lambda \delta_{\boldsymbol{W}_L}$
  $\boldsymbol{W}_1^+, \ldots, \boldsymbol{W}_L^+ \leftarrow \boldsymbol{W}_1^+ - \lambda \delta_{\boldsymbol{W}_1^+}, \ldots, \boldsymbol{W}_L^+ - \lambda \delta_{\boldsymbol{W}_L^+}$
**end for**

---

