# OpenReview forum: "A biologically-plausible alternative to backpropagation using pseudoinverse feedback"
_ICLR.cc/2025/Conference — Submitted to ICLR 2025_

### Official Review · Reviewer_NytU · 2024-10-21

**Soundness:** 2
**Presentation:** 3
**Contribution:** 1
**Rating:** 3
**Confidence:** 5

**Summary:**

This work presents an alternative to weight-transpose based measurement of gradients for updating and optimization of neural network models. Specifically, a method for the use of (pseudo)inverse-based top-down models of each layer of a network’s activity is described. A ‘sleep-wake’-esque cycle of updating is used to compute the Moore-Penrose pseudoinverse of weight matrices in deep neural network architectures, with a relation to the recirculation algorithm. Thereafter, the Hildebrandt-Graves Theorem is applied to show how such pseudoinverses can provide feedback to layers of a neural network for an alternative optimization method of neural networks. This is applied to the MNIST and CIFAR-10 classification tasks to demonstrate its performance.

**Strengths:**

- The work has a well written narrative structure. A reader is led well through the method as well as through the steps for running this method. Algorithms and mathematical steps are relatively clear.
- The relation of inverse-based credit assignment to the Hildebrandt-Graves Theorem is original (novel) and can contribute to the discussion around the plausibility or efficacy of target-propagation methods.
- Illustrations are used effectively to describe the process of computing the proposed inversion and forward parameter learning processes.

**Weaknesses:**

- This work has not been embedded within recent literature. This is a major drawback as it misses out the ways in which existing research has contributed to this question and does not provide comparison (theoretical or empirical) against these methods. There exist works which describe: how target propagation relates to iterative approximate inverses (Bengio 2020), how to dynamically compute the Moore-Penrose inverse (Podlaski et al 2020), how the use of inverses relates to second order optimization (Meulemans et al. 2020), and how in special cases the inverse and transpose are equivalent (Ahmad et al. 2020). These are just some of the recent works which have contributed to this story and are missing in this context, many of which also fulfil some of the theoretical requirements outlined in this work.
- The experiments presented are much too limited to draw conclusions from. In the results shown, simulations are extremely short (in terms of epochs), do not have multiple repeats, and are within a relatively low task complexity range. Existing work has often indicated that traditional target propagation can fail suddenly, and not scale to tasks of greater difficulty than MNIST or CIFAR-10 (See Bartunov et al. 2018). Furthermore, the results reached by backpropagation are far far short of what is possible with longer training and coupling with adaptive optimizers. To prove that these results are robust and scale, one would desire multiple repeats, training until convergence (100+ epochs), and application to a more challenging task. Furthermore, one would expect comparison against traditional target propagation or difference target propagation to see whether this method is comparable or worse.
- The computational complexity and robustness of hyperparameters is not much discussed but would be important for recreation of results. A hyperparameters table is missing for a reader to understand exactly how the parameters were tuned.
- There are some minor textual issues, for example feedback alignment is referred to as ‘random target propagation’ in the abstract (a term that this reviewer has never encountered) but later referred to as ‘feedback alignment’ as it is called in the original referenced paper.


Bengio, Y. (2020). Deriving Differential Target Propagation from Iterating Approximate Inverses. In arXiv [cs.LG]. arXiv. http://arxiv.org/abs/2007.15139

Podlaski, W. F., & Machens, C. K. (2020). Biological credit assignment through dynamic inversion of feedforward networks. In arXiv [q-bio.NC]. arXiv. http://arxiv.org/abs/2007.05112 (Published in Neurips 2020)

Meulemans, A., Carzaniga, F. S., Suykens, J. A. K., Sacramento, J., & Grewe, B. F. (2020). A Theoretical Framework for Target Propagation. In arXiv [cs.LG]. arXiv. http://arxiv.org/abs/2006.14331 (Published Neurips 2020)

Ahmad, N., van Gerven, M. A. J., & Ambrogioni, L. (2020). GAIT-prop: A biologically plausible learning rule derived from backpropagation of error. In arXiv [cs.LG]. arXiv. https://arxiv.org/abs/2006.06438 (Published Neurips 2020)

Bartunov, S., Santoro, A., Richards, B. A., Marris, L., Hinton, G. E., & Lillicrap, T. (2018). Assessing the Scalability of Biologically-Motivated Deep Learning Algorithms and Architectures. In arXiv [cs.LG]. arXiv. http://arxiv.org/abs/1807.04587 (Published Neurips 2018)

**Questions:**

I have no further questions and would point to the weaknesses section for a full list of actionable critiques. The review period is rather short to correct such a list of weaknesses. Nonetheless, should all of these points be addressed sufficiently I would be willing to revise my score.

---

> ### Author Response · Authors · 2024-11-27
>
> **W1**: Insufficiently contextualized within recent literature
>
> Thank you for bringing these papers to our attention.
>
> Our algorithm is similar to that proposed by Meuleumans et. al (2020), and we identify the same problems as them in connecting target propagation to Gauss-Newton optimization. More specifically, we both identified the problem that the pseudoinverse of the whole network can’t be factorized as the pseudoinverse of each individual layer.
>
> However, our solution differs from Meulemans et. al (2020), in that **we do not use direct feedback connections** from the final layer to each intermediate layer. Our method preserves the modular, stacked-autoencoder structure proposed under the target-propagation framework, and uses a Newton-like optimization method which **only requires local, layer-wise feedback connections**. Overall, we believe that our proposal constitutes a different theoretical framework for target-propagation-like methods, which has greater biological-plausibility than Meulemans et. al. (2020).
>
> Ahmad et. al (2020) and Bengio et. al (2020) also propose similar methods to ours, with the use of layer-wise inverses to propagate error. However, the mathematical derivations in Ahmad et. al (2020) and Bengio et. al (2020) both assume that the network is invertible. In that case, both our method, GAIT-prop and target-propagation are equivalent to Gauss-Newton optimization. Our method differs from these previous approaches in that **we do not require perfect invertibility** - allowing for greater architectural flexibility.
>
> Overall, our method can be considered an additional mathematical framework by which the methods in these papers can be better understood and generalized.
>
> **W2**: Experiments are too limited
>
> We agree that the current set of experiments are limited, however, we think that they constitute a sufficient proof-of-concept for a new optimization method, which we have validated mathematically.
>
> Furthermore, in our updated draft, we will include multiple seed initializations and run each method for more epochs.
>
> **W3**: Hyperparameters are not included
>
> Thank you for the suggestion. We have included a table of hyperparameters in our latest draft

---

> > ### Comment · Reviewer_NytU · 2024-11-28
> >
> > Thank you for your response.
> >
> > __W1__: Insufficiently contextualized within recent literature
> >
> > The embedding in the literature has improved somewhat. However, missing is still any reference to how your method (which learns the Moore-Penrose inverse) is similar to/alternative to the approach of Podlaski et al. 2020 (Biological Credit Assignment through Dynamic Inversion of Feedforward Networks) which dynamically produces the same pseudo-inverse. Note also some misconceptions in your statements: GAIT-prop in fact shows equivalence to backpropagation and not Gauss-Newton optimization.
> >
> > Having said that, the framework of your paper is indeed a useful contribution, though weaknesses remain.
> >
> > __W2__: Experiments are too limited
> >
> > Experiments are limited but also in my opinion far far away from being convincing. Even considering the current results as they are, BP show extremely low accuracy at even MNIST. As an example, in the Podlaski paper they were able to train BP to a test error of 2.2% and target propagation (with dynamic inversion) to an error of 2.8%. This is far away from the >7% error rate that you show for all algorithms. And in their case it was a simple two layer network (784-1000-10 nodes). I believe that the results at present are misleading compared to similar existing literature. I would even be happy to accept the current scale of the tasks if they were carried out more robustly and for CIFAR-10 with a proper data augmentation (which I assume is missing given it's low accuracy).
> >
> > Furthermore, note that there are a few errors in the paper at present: I believe your headings in Figure 4 are the wrong way around.
> >
> > __W3__: Hyperparameters are not included
> >
> > Thank you for their inclusion.

---

### Official Review · Reviewer_ujbb · 2024-10-31

**Soundness:** 2
**Presentation:** 3
**Contribution:** 2
**Rating:** 5
**Confidence:** 4

**Summary:**

This work aims to describe a biologically plausible alternative to the backpropagation of error algorithm. The authors contend that algorithms sending predictions to lower layers are more biologically plausible than those that depend on feedback error networks. Their algorithm, called reciprocal feedback learning, which falls in the former category, involves a biologically plausible mechanism (i.e., local numerical integration) to compute the pseudoinverse of the weights of a layer. This mechanism is an extension of recirculation algorithms. They then use the Hildebrandt-Graves Theorem to develop an algorithm to propagate errors in the backward pass of the network. The specific claim of the paper is that this algorithm avoids the biological implausibility of weight transport by circumventing explicit calculation of a weight transpose. On MNIST and CIFAR, they show that the algorithm has comparable asymptotic performance to backpropagation and converges in fewer iterations than conventional random target propagation algorithms. The contribution is a comprehensive derivation of a novel algorithm to calculate the pseudoinverse, a novel training algorithm, positive evidence from benchmark comparisons, and an overall claim of biological plausibility compared with other algorithms.

**Strengths:**

The idea is interesting and clearly and concisely presented. The algorithm is thoroughly derived and described with thoughtful attention to detail. The work makes some nice connections between previously disparate algorithms and theorems. It adds a novel idea to the growing research on alternative backpropagation, particularly in algorithms that address the weight transport problem.

**Weaknesses:**

The work is poorly situated in the existing literature on biologically plausible alternatives. Thus, a main weakness of the work is the insufficient comparison and benchmarking against other algorithms. In particular, the biological motivation is rather weak, especially considering that the authors recognize that predictive coding algorithms have strong biological underpinnings. While it is true that predictive coding (PC) suffers from the weight transport problem, there have been works showing it is also robust to randomness in backward connections (see https://arxiv.org/pdf/2010.01047). I would expect a comparison against these algorithms, especially when weight transport assumptions are relaxed.

It is true that PC has not been scaled to problems on the scale of ImageNet. If the argument here is that this algorithm comes from a family of algorithms that have been shown to scale, there should be an attempt to test it on ImageNet. More generally, the algorithm is only tested against an early algorithm from this family—random feedback—rather than newer algorithms like sign symmetry.

Overall, it is not clear what merit this algorithm has over existing literature. From an engineering perspective, it is another compute-heavy alternative to backpropagation, and it is not clear how it is better than other algorithms. For example, it would be good to compare the computational expense (how many pseudoinverse iterations per learning update are needed and the number of learning updates required to achieve asymptotic performance) and accuracy to BP, PC, target propagation, etc.

Its alignment with neuroscience appears weak, and methods like PC have much greater resonance with neural circuitry. While this algorithm solves the weight transport issue for PC, its biological potential seems limited. Until a closer analysis of possible underlying neural circuitry is conducted, or at least a direct comparison with PC when weight transport assumptions are relaxed, its biological plausibility remains uncertain.

**Questions:**

One of the biggest barriers to the adoption of alternatives to backpropagation is increased computational expense. Reducing this is a key focus of the biomorphic algorithms community, and a comparison against the computational expense of other similar algorithms is needed. For example, it is known that PC algorithms can be trained with roughly 2N iterations, whereas this method requires at least 60. How robust is the algorithm if the pseudoinverses do not fully converge?
While the description of the system as a sleep-wake algorithm is interesting, I am unsure of the biological plausibility. Aren't you limited to a single update during the wake phase, after which you would have to learn another pseudoinverse? Similarly, how robust is the algorithm to performing multiple weight updates in the wake phase? For small learning rates, might there be some tolerance? These aspects should be tested.

Relatedly, I assume that recalculating the pseudoinverse after one step of learning might actually be inexpensive, i.e., the original pseudoinverse could provide a good initialization.

In the discussion, the paper states that second-order, Newton-like methods tend to converge faster and to flatter minima than gradient descent, but they cite only one paper as evidence. Is this always true? Could you provide additional references to support this claim, clarify under what conditions it holds, or be more conservative with this claim?

Another concern I have is that (unlike target propagation) the error signal is backpropagated from the output layer all the way through the network. Thus, it is not clear if this computation is completely parallel, as in other algorithms (e.g., PC). Doesn't this jeopardize the biological plausibility?

---

> ### Author Response · Authors · 2024-11-27
>
> **W1:** Issues with the sleep/wake training scheme
>
> Thank you for your suggestions. In the latest draft, we have included simulations where forward and feedback weights are trained concurrently utilizing an iteratively updated approximate pseudoinverse at each training step. This method has a lower computational complexity, but has been less rigorously studied.
>
> Under the sleep-wake training scheme, we assume several steps are made during each phase, with a small enough learning rate such that the feedback weights learned during the previous “sleep” period are close enough to the true pseudoinverse.
>
>
> **W2:** Convergence speed of second-order optimization
>
> Thank you for pointing this out. When the loss landscape is relatively convex, it is theoretically true that second-order methods will tend to converge faster than first-order gradient descent, since each step size is “scaled” by the inverse of the Hessian matrix. Practical approximations of second-order optimization, such as K-FAC (Martens and Grosse 2015) and Shampoo (Gupta, et. al 2018), tend to converge faster than gradient descent, usually at the cost of additional computational complexity.
>
> In regards to convergence to flatter minima, we acknowledge that there is not enough experimental evidence to support this, and we will remove that statement in our future draft.
>
>
> **W3:** Biological implausibility in comparison to predictive coding
>
> From our understanding, the family of models underthe predictive coding framework have included stacked-autoencoder architectures similar to ours (such as Marino 2020, and  Whittington and Bogatz 2017). However, we acknowledge that our architecture is more simplified than those referenced, as do not include recurrent lateral connections, or error-encoding neurons.
>
> In regards to directly comparing with PC, we cannot compare our method to PC with weight constraints relaxed, as our method is a proposed solution to the weight transport problem itself. In fact, when benchmarked against random feedback weights (Feedback Alignment), our method converges faster. **Our algorithm is not intended as a direct alternative to energy-based PC architectures, such as that of Millidge et. al 2020, but rather a solution to the weight transport problem encountered by many such models** (inspired by the PC framework).
>
> **Q1:**
> Could you please clarify what you mean by the algorithm being completely parallel? Is that in reference to having separate sleep/wake training phases?
>
> James C. R. Whittington, Rafal Bogacz; An Approximation of the Error Backpropagation Algorithm in a Predictive Coding Network with Local Hebbian Synaptic Plasticity. Neural Comput 2017; 29 (5): 1229–1262. doi: https://doi.org/10.1162/NECO_a_00949
>
> Marino, J. (2021). Predictive Coding, Variational Autoencoders, and Biological Connections. arXiv [Cs.NE]. http://arxiv.org/abs/2011.07464
>
> Gupta, V., Koren, T., & Singer, Y. (2018). Shampoo: Preconditioned Stochastic Tensor Optimization. arXiv [Cs.LG]. http://arxiv.org/abs/1802.09568
>
> Martens, J., & Grosse, R. B. (2015). Optimizing Neural Networks with Kronecker-factored Approximate Curvature. CoRR. http://arxiv.org/abs/1503.05671

---

> > ### Comment · Reviewer_ujbb · 2024-11-28
> > **Nice proof of concept but needs a more detailed quantitative and qualitative comparison to other algorithms**
> >
> > Thanks for the response. It is nice to see there is evidence that concurrent training, rather than the sleep-wake conceptualization, is more compute efficient, but ideally, this does need to be rigorously studied to augment the overall contribution.
> >
> > On the parallel nature: I mean that in PC, inference and learning are completely parallelized, i.e., there is no such thing as a backward or forward pass. I think this algorithm, if I’m not mistaken, has an effective backward pass during learning, requiring coordination of how the information flows.
> >
> > While I appreciate the extra results, I still feel that comparison to other algorithms has not been comprehensive, and thus it’s not clear whether there are significant contributions to biological plausibility or engineering. Relaxing the weight-transpose problem is interesting, but it also introduces a suite of other commitments that  plausibility needs to be argued for. As an engineering paradigm, it is not clear how computationally expensive it is compared to other algorithms. I agree this is a nice proof of concept, but without more comprehensive investigation and comparison of these ideas to other works I will maintain my already relatively high score.

---

### Official Review · Reviewer_wCKF · 2024-11-04

**Soundness:** 3
**Presentation:** 2
**Contribution:** 3
**Rating:** 5
**Confidence:** 3

**Summary:**

In this paper, the authors propose a mechanism to train deep neural networks that solves some of the biological implausible aspects of back propagation. In particular, they propose Reciprocal Feedback:
* The propose a modification of the Recirculation algorithm that can learn the Moore-Penrose pseudo-inverse of the feedforward (or feedback) weights
* Using the Hildenbrandt-Graves Theorem they show that the learned pseudo inverse can be used as an alternative to traditional gradient descent (which relies on the transpose of the feedforward weights)

They show some preliminary result on the Mnist and Cifar10 classification tasks, and compare them with Backpropagation and biologically plausible algorithms.

**Strengths:**

I believe the paper is well structured and shows interesting methods and results. In detail:
* The theory part is well explained and are very complete and self contained.
* The results and dynamics of the algorithms are well explained by the theory.
* The assumptions needed for the algorithm to work are clearly stated.

**Weaknesses:**

I believe this paper is solid, but I will give it a 5 because of the insufficient contextualization with the previous literature. I would be happy to improve my score if these concerns are addressed:
* Section 2 (Related work) is very brief and it is not enough for readers who are not familiar with the mentioned algorithms. For example, a more detailed explanation of Target Propagation, Recirculation Algorithm and Weight-Mirroring would be useful to make the paper more clear.
* I like the fact that the authors jump straight into the theory, but I believe that I could be better contextualized by comparing the results with the Related Work section algorithms.
* I believe the results on the MNIST on CIFAR10 are missing some key details: how were the hyperparameters chosen? I find the results for backpropagation surprisingly worse than what I would expect (e.g, >97% for MNIST)

In general, although the paper is well written, I believe space could be optimized to include these suggestions, in particular a more complete explanation of the related work and a comparison, which are needed to contextualize the paper.

**Questions:**

* It is not super clear to me how this algorithm compares to others in terms of computational complexity, it would be nice to see it as a metric of comparison
* The word biologically plausible is used a lot, and I do agree that this algorithm may solve some of the issues of backpropagation (e.g. weight simmetry), but there is no mention on how it could be implemented in biological networks. Could you elaborate more on that?

---

### Meta-Review · Area_Chair_KcXJ · 2024-12-19

**Metareview:**

The authors propose a novel optimization method, Reciprocal Feedback Learning, which aims to address the biological implausibility of backpropagation. Preliminary results are presented on MNIST and CIFAR-10, comparing the proposed approach with backpropagation and other biologically inspired methods.

While the paper provides a solid theoretical foundation and an interesting contribution to biologically plausible learning, several issues were identified by the reviewers, particularly in terms of contextualization, experimental rigor, and comparison with prior work. All three reviewers rated the submission as below the acceptance threshold, citing overlapping concerns.

**Additional Comments On Reviewer Discussion:**

The authors made some improvements during the rebuttal period, addressing contextualization, experiments, and minor presentation issues, but the responses fell short of fully addressing the reviewers’ major concerns.

---

### Decision · Program_Chairs · 2025-01-22

Reject